# Towards counteracting adversarial perturbations to resist adversarial examples

## Abstract

Studies show that neural networks are susceptible to adversarial attacks. This exposes a potential threat to neural network-based artificial intelligence systems. We observe that the probability of the correct result outputted by the neural network increases by applying small perturbations generated for non-predicted class labels to adversarial examples. Based on this observation, we propose a method of counteracting adversarial perturbations to resist adversarial examples. In our method, we randomly select a number of class labels and generate small perturbations for these selected labels. The generated perturbations are added together and then clamped onto a specified space. The obtained perturbation is finally added to the adversarial example to counteract the adversarial perturbation contained in the example. The proposed method is applied at inference time and does not require retraining or finetuning the model. We validate the proposed method on CIFAR-10 and CIFAR-100. The experimental results demonstrate that our method effectively improves the defense performance of the baseline methods, especially against strong adversarial examples generated using more iterations.

## 1 Introduction

Deep neural networks (DNNs) have become the dominant approach for various tasks including image understanding, natural language processing and speech recognition (He et al., 2016; Devlin et al., 2018; Park et al., 2018). However, recent studies demonstrate that neural networks are vulnerable to adversarial examples (Szegedy et al., 2014; Goodfellow et al., 2015). That is, these network models make an incorrect prediction with high confidence for inputs that are only slightly different from correctly predicted examples. This reveals a potential threat to neural network-based artificial intelligence systems, many of which have been widely deployed in real-world applications.

The adversarial vulnerability of neural networks reveals fundamental blind spots in the learning algorithms. Even with advanced learning and regularization techniques, neural networks are not learning the true underlying distribution of the training data, although they can obtain extraordinary performance on test sets. This phenomenon is now attracting much research attention. There have been increasing studies attempting to explain neural networks' adversarial vulnerability and develop methods to resist adversarial examples (Madry et al., 2018; Zhang et al., 2020; Pang et al., 2020). While much progress has been made, most existing studies remain preliminary. Because it is difficult to construct a theoretical model to explain the adversarial perturbation generating process, defending against adversarial attacks is still a challenging task.

Existing methods of resisting adversarial perturbations perform defense either at training time or inference time. Training time defense methods attempt to increase model capacity to improve adversarial robustness. One of the commonly used methods is adversarial training (Szegedy et al., 2014), in which a mixture of adversarial and clean examples are used to train the neural network. The adversarial training method can be seen as minimizing the worst case loss when the training example is perturbed by an adversary (Goodfellow et al., 2015). Adversarial training requires an adversary to generate adversarial examples in the training procedure. This can significantly increase the training time. Adversarial training also results in reduced performance on clean examples. Lamb et al. (2019) recently introduced interpolated adversarial training (IAT) that incorporates interpolation-based training into the adversarial training framework. The IAT method helps to improve performance on clean examples while maintaining adversarial robustness.

As to inference time defense methods, the main idea is to transfer adversarial perturbations such that the obtained inputs are no longer adversarial. Tabacof & Valle (2016) studied the use of random noise such as Gaussian noise and heavy-tail noise to resist adversarial perturbations. Xie et al. (2018) introduced to apply two randomization operations, *i.e.,* random resizing and random zero padding, to inputs to improve adversarial robustness. Guo et al. (2018) investigated the use of random cropping and rescaling to transfer adversarial perturbations. More recently, Pang et al. (2020) proposed the mixup inference method that uses the interpolation between the input and a randomly selected clean image for inference. This method can shrink adversarial perturbations somewhat by the interpolation operation. Inference time defense methods can be directly applied to off-the-shelf network models without retraining or finetuning them. This can be much efficient as compared to training time defense methods.

Though adversarial perturbations are not readily perceivable by a human observer, it is suggested that adversarial examples are outside the natural image manifold (Hu et al., 2019). Previous studies have suggested that adversarial vulnerability is caused by the locally unstable behavior of classifiers on data manifolds (Fawzi et al., 2016; Pang et al., 2018). Pang et al. (2020) also suggested that adversarial perturbations have the locality property and could be resisted by breaking the locality. Existing inference time defense methods mainly use stochastic transformations such as mixup and random cropping and rescaling to break the locality. In this research, we observe that applying small perturbations generated for non-predicted class labels to the adversarial example helps to counteract the adversarial effect. Motivated by this observation, we propose a method that employs the use of small perturbations to counteract adversarial perturbations. In the proposed method, we generate small perturbation using local first-order gradient information for a number of randomly selected class lables. These small perturbations are added together and projected onto a specified space before finally applying to the adversarial example. Our method can be used as a preliminary step before applying existing inference time defense methods.

To the best of our knowledge, this is the first research on using local first-order gradient information to resist adversarial perturbations. Successful attack methods such as projected gradient descent (PGD) (Madry et al., 2018) usually use local gradient to obtain adversarial perturbations. Compared to random transformations, it would be more effective to use local gradient to resist adversarial perturbations. We show through experiments that our method is effective and complementary to random transformation-based methods to improve defense performance.

The contributions of this paper can be summarized as follows:

- We propose a method that uses small first-order perturbations to defend against adversarial attacks. We show that our method is effective in counteracting adversarial perturbations and improving adversarial robustness.

- We evaluate our method on CIFAR-10 and CIFAR-100 against PGD attacks in different settings. The experimental results demonstrate that our method significantly improves the defense performance of the baseline methods against both untargeted and targeted attacks and that it performs well in resisting strong adversarial examples generated using more iterations.

## 2 PRELIMINARY

### 2.1 ADVERSARIAL EXAMPLES

We consider a neural network $f(\cdot)$ with parameters $\boldsymbol{\theta}$ that outputs a vector of probabilities for $L = \{1, 2, ..., l\}$ categories. In supervised learning, empirical risk minimization (ERM) (Vapnik, 1998) has been commonly used as the principle to optimize the parameters on a training set. Given an input $\boldsymbol{x}$, the neural network makes a prediction $c(\boldsymbol{x}) = \arg\max_{j \in L} f_j(\boldsymbol{x})$. The prediction is correct if $c(\boldsymbol{x})$ is the same as the actual target $c^*(\boldsymbol{x})$.

Unfortunately, ERM trained neural networks are vulnerable to adversarial examples, inputs formed by applying small but intentionally crafted perturbations (Szegedy et al., 2014; Madry et al., 2018). That is, an adversarial example $\boldsymbol{x}'$ is close to a clean example $\boldsymbol{x}$ under a distance metric, *e.g.,* $\ell_\infty$ distance, but the neural network outputs an incorrect result for the adversarial example $\boldsymbol{x}'$ with high

confidence. In most cases, the difference between the adversarial example and clean example is not readily recognizable to humans.

## 2.2 ATTACK METHODS

Existing attack methods can be categorized into white-box attacks and black-box attacks. We focus on defending against white-box attacks, wherein the adversary has full access to the network model including the architecture and weights. The fast gradient sign (FGSM) method (Goodfellow et al., 2015) and PGD are two successful optimization-based attack methods.

**The FGSM method** is a one-step attack method. It generates adversarial perturbations that yield the highest loss increase in the gradient sign direction. Let $x$ be the input to a network model, $y$ the label associate with $x$ and $L(\theta, x, y)$ be the loss function for training the neural network. The FGSM method generates a max-norm constrained perturbation as follows:

$$\eta = \varepsilon \text{sign}(\nabla_x L(\theta, x, y)), \tag{1}$$

where $\varepsilon$ denotes the max-norm. This method was developed based on the view that the primary cause of neural networks' adversarial vulnerability is their linear nature. The required gradient can be computed efficiently using backpropagation.

**The PGD method** is a multistep attack method that iteratively applies projected gradient descent on the negative loss function (Kurakin et al., 2016) as follows:

$$x^{t+1} = \Pi_{x+\mathcal{S}}(x^t + \alpha \text{sign}(\nabla_{x^t} L(\theta, x^t, y))), \tag{2}$$

where $\alpha$ denotes the step size and $\Pi$ denotes the projection operator that projects the perturbed input onto $x + \mathcal{S}$. We consider projecting the perturbed input onto a predefined $\ell_\infty$ ball from the original input. The PGD attack method can be seen as a multistep FGSM method. It is a much strong adversary that reliably causes a variety of neural networks to misclassify their input.

## 3 METHODOLOGY

While many studies have been conducted on defending against adversarial attacks at inference time, these studies have not considered using local gradient information to resist adversarial perturbations. Previous work has suggested that the primary cause of neural networks' adversarial vulnerability is their linear nature (Goodfellow et al., 2015). It would be more effective to use first-order gradient information to counteract adversarial perturbations such that the resulted perturbations no longer result in the model making an incorrect prediction.

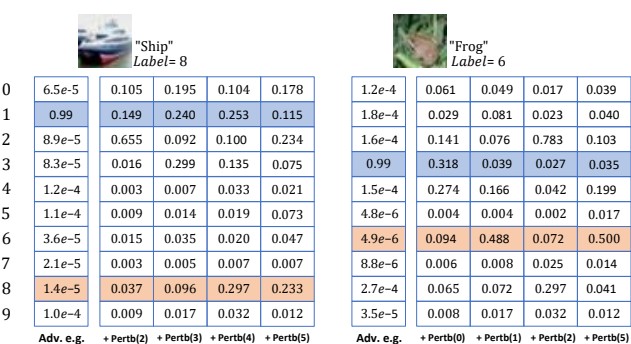

Predicted probabilities outputted by ResNet50. Adversarial examples are crafted using the untargeted $\text{PGD}_{10}$ method on images from CIFAR-10.

Figure 1: An illustration that shows applying small perturbations generated for non-predicted labels to the adversarial example helps suppress the adversarial effect and improve the prediction probability for the correct category. +Pertb(i) denotes applying the small perturbation generated for $i$-th class label to the adversarial example.

Adversarial perturbations are small crafted perturbations that slightly affect the visual quality of inputs but cause the neural network to misclassify the inputs in favor of an incorrect answer with high probability. We show that this effect can be counteracted by applying small perturbations generated using local first-order gradient information for class labels other than the predicted one. An illustration of this phenomenon is shown in Figure 1. We see that by adding perturbations generated for non-predicted labels to the input, the prediction probability for the correct category increases and that for the incorrect label is suppressed.

---

**Algorithm 1** Counteracting adversarial perturbations using local first-order gradient.

---

**Input:** Neural network $f$; input $\boldsymbol{x}$; step size $\alpha$ used in PGD to generate perturbations to counteract the adversarial perturbation.
**Output:** Prediction result for $\boldsymbol{x}$.
1: Randomly select $N$ class labels $\{l_1, l_2, ..., l_N\}$;
2: **for** $i = 1$ to $N$ **do**
3:     $\eta_i = PGD(l_i, \alpha, \text{step}=1)$ // generate perturbation $\eta_i$ for $l_i$ using the one-step PGD method.
4: **end for**
5: $\boldsymbol{x} = \boldsymbol{x} + \Pi_{\mathcal{C}}(\sum_{i=1}^{N} \eta_i(\boldsymbol{x}))$ // $\mathcal{C}$ is a $\ell_\infty$ bounded space.
6: **return** $f(\boldsymbol{x})$.

---

Based on this phenomenon, we propose a method of counteracting adversarial perturbations to improve adversarial robustness. In the proposed method, we generate small perturbations for a number of randomly selected class labels and apply these perturbations to the input to resist the adversarial perturbation. Let $\boldsymbol{x}$ be the input to a model, which can be an adversarial or clean example. We randomly select $N$ class labels and generate small first-order perturbations for the $N$ selected labels. These $N$ small perturbations are added together and then projected onto a $\ell_\infty$-bounded space before applying to the input. This procedure can be formulated as follows:

$$\tilde{\boldsymbol{x}} = \boldsymbol{x} + \Pi_{\mathcal{C}}(\sum_{i=1}^{N} \eta_i(\boldsymbol{x})), \tag{3}$$

where $\eta_i(\boldsymbol{x})$ denotes the small perturbation generated for the $i$-th selected class label, $\mathcal{C} = \{\boldsymbol{t} \mid \|\boldsymbol{t} - \boldsymbol{x}\|_\infty \leq \mu\}$ is a $\mu$ bounded $\ell_\infty$ space. The one-step PGD method is used to generate small perturbations. This is the same as using the FGSM method and empirically achieves better performance than using multiple steps. The perturbations can be generated in an untargeted or targeted manner. The combined perturbation is projected onto the space $\mathcal{C}$. This ensures that the obtained example is visually similar to the original one. We detail the procedure for counteracting adversarial perturbations in Algorithm 1.

**Discussion and Analysis** Adversarial examples exposes underlying flaws in the training algorithms. While much progress has been made in defending against adversarial attacks, it is difficult to theoretically understand neural networks' vulnerability to adversarial examples. Previous work (Athalye et al., 2018) has suggested that the adversarial perturbation $\delta$ can be obtained by solving the following optimization problem:

$$\min \|\delta\|_p,$$
$$s.t. \ c(\boldsymbol{x} + \delta) \neq c^*(\boldsymbol{x}), \|\delta\|_p \leq \xi, \tag{4}$$

where $\xi$ is a hyperparameter constraining the size of the perturbation. This problem can be effectively solved by gradient descent-based attack methods such as PGD and FGSM that reliably cause neural networks to output an incorrect result. These attack methods typically use local first-order gradient to find the optimal solution. Because state-of-the-art neural networks usually have many parameters, perturbations obtained with these attack methods may overfit to the inputs. Therefore, perturbing and transferring these adversarial perturbations could be an effective way to resist the adversarial effect. Unlike previous random transformation-based methods, we employ the use of local first-order gradient information to counteract the adversarial effect. We show that the proposed method is effective in improving defense performance, especially against strong adversarial examples generated using more iterations.

Let $\boldsymbol{x}_0$ be a clean example and $\delta$ be the adversarial perturbation. In our method, the following input is fed to the neural network:

$$\boldsymbol{x}_0 + \delta \cdot \mathbf{1}_{z(\boldsymbol{x}_0)} + \Pi_{\mathcal{C}}(\sum_{i=1}^{N} \eta_i(\boldsymbol{x}_0)), \text{ where } \mathbf{1}_{z(\boldsymbol{x}_0)} = \begin{cases} 0, & \boldsymbol{x}_0 \text{ is not subject to adversarial attack,} \\ 1, & \boldsymbol{x}_0 \text{ is subject to adversarial attack.} \end{cases}$$
$$\tag{5}$$

The perturbation $\eta_i$ generated to counteract the adversarial perturbation should be small, otherwise it would be a new adversarial perturbation. This would essentially have no effect in counteracting the

Table 1: Classification accuray (%) on adversarial exmples crafted on the test set of CIFAR-10. We report performance of resisting PGD attacks with 10, 50 and 200 iterations. The numbers in parentheses are performance improvements achieved by applying our method.

| Method | Untargeted attacks | | | Targeted attacks | | |
|---|---|---|---|---|---|---|
| | $PGD_{10}$ | $PGD_{50}$ | $PGD_{200}$ | $PGD_{10}$ | $PGD_{50}$ | $PGD_{200}$ |
| Mixup (Zhang et al., 2018) | 3.6 | 3.2 | 3.1 | $\leq 1$ | $\leq 1$ | $\leq 1$ |
| Ours + Mixup | 26.5 | 27.2 | 31.7 | 55.3 | 59.7 | 63.1 |
| Xie et al.'s (2018) + Mixup | 23.0 | 19.6 | 19.1 | 38.4 | 31.1 | 25.2 |
| Ours + Xie et al.'s + Mixup | $35.5_{(+12.5)}$ | $36.4_{(+16.8)}$ | $41.3_{(+22.2)}$ | $60.3_{(+21.9)}$ | $63.7_{(+32.6)}$ | $66.1_{(+40.9)}$ |
| Guo et al.'s (2018) + Mixup | 31.2 | 28.8 | 28.3 | 57.8 | 49.1 | 48.9 |
| Ours + Guo et al.'s + Mixup | $\mathbf{50.5}_{(+19.3)}$ | $\mathbf{51.8}_{(+23.0)}$ | $\mathbf{56.2}_{(+27.9)}$ | $\mathbf{74.1}_{(+16.3)}$ | $\mathbf{78.0}_{(+28.9)}$ | $\mathbf{79.8}_{(+30.9)}$ |
| MI-OL (Pang et al., 2020) + Mixup | 26.1 | 18.8 | 18.3 | 55.6 | 51.2 | 50.8 |
| Ours + MI-OL + Mixup | $38.8_{(+12.7)}$ | $38.6_{(+19.8)}$ | $41.5_{(+23.2)}$ | $55.3_{(-0.3)}$ | $60.0_{(+8.8)}$ | $63.0_{(+12.2)}$ |
| IAT (Lamb et al., 2019) | 46.7 | 43.5 | 42.5 | 65.6 | 62.5 | 61.9 |
| Ours + IAT | 60.1 | 60.8 | 60.9 | 68.0 | 69.1 | 68.9 |
| Xie et al. (2018) + IAT | 59.7 | 58.4 | 57.9 | 71.1 | 69.7 | 69.3 |
| Ours + Xie et al.'s + IAT | $\mathbf{68.7}_{(+9.4)}$ | $\mathbf{69.2}_{(+10.8)}$ | $\mathbf{69.3}_{(+11.4)}$ | $\mathbf{76.7}_{(+5.6)}$ | $\mathbf{76.6}_{(+6.9)}$ | $\mathbf{76.6}_{(+7.3)}$ |
| Guo et al.'s (2018) + IAT | 60.9 | 60.7 | 60.3 | 73.2 | 72.1 | 71.6 |
| Ours + Guo et al.'s + IAT | $67.2_{(+6.3)}$ | $68.1_{(+7.4)}$ | $68.2_{(+7.9)}$ | $72.6_{(-0.6)}$ | $72.4_{(+0.3)}$ | $72.8_{(+1.2)}$ |
| MI-OL (Pang et al., 2020) + IAT | 64.5 | 63.8 | 63.3 | 75.3 | 74.7 | 74.5 |
| Ours + MI-OL + IAT | $68.6_{(+4.1)}$ | $68.9_{(+5.1)}$ | $68.8_{(+5.5)}$ | $72.4_{(-2.9)}$ | $72.8_{(-2.9)}$ | $72.9_{(-1.6)}$ |

adversarial perturbation. Adversarial training that has been shown to be effective to improve adversarial robustness usually employs a first-order adversarial like PGD to provide adversarial examples for training. These adversarial examples help to regularize the model to be resistant to adversarial perturbations. We show through experiments that our method is complementary to adversarial training to improve overall defense performance against both untargeted and targeted attacks.

The proposed method is applied at inference time. It can be directly applied to off-the-shelf models without retraining or finetuning them. The required gradient for generating small perturbations can be computed efficiently in parallel using backpropagation. This would not increase too much time for inference.

## 4 EXPERIMENTS

### 4.1 EXPERIMENTAL SETUP

We conduct experiments on CIFAR-10 and CIFAR-100 (Krizhevsky et al., 2009). ResNet-50 (He et al., 2016) is used as the network model. We validate the proposed method on models trained using two methods: Mixup (Zhang et al., 2018) and IAT (Lamb et al., 2019). For fair performance comparison, we follow the same experimental setup as Pang et al. (2020) to train the models. The training procedure is performed for 200 epochs with a batch size of 64. The learning rate is initialized to 0.1 and divided by a factor of 10 at epoch 100 and 150. The values used for interpolation are sampled from $Beta(1, 1)$ for both Mixup and IAT. The ratio between clean examples and adversarial examples used in IAT is set to 1:1. The untargeted $PGD_{10}$ method with a step size of $2/255$ and $\varepsilon$ set to $8/255$ is used to generate adversarial examples in IAT.

We experiment against both untargeted and targeted PGD attacks with different iterations. The values of $\varepsilon$ and step size for the PGD attacks are set to $8/255$ and $2/255$, respectively. The one-step PDG method is used to generate perturbations to resist adversarial perturbations. Unless stated otherwise, perturbations used for defense purposes are generated in a targeted fashion. The step size for the one-step PGD and number of randomly selected class labels are set to $4/255$ and 9, respectively. The value of $\mu$ is set to $8/255$. For each experiment, we run our model for three times and report the mean accuracy. Our method is implemented in Pytorch (Paszke et al., 2017) and all experiments are conducted on one GPU.

**Baselines** Three methods that were recently developed for inference time defense are used as baselines. These three methods are Xie et al.'s (2018), Guo et al.'s (2018) and MI-OL (mixup inference

Table 2: Classification accuray (%) on adversarial examples crafted on the test set of CIFAR-100. We report performance of resisting PGD attacks with 10, 50 and 200 iterations. For targeted attacks on IAT models, defense perturbations are generated in an untargeted manner. The numbres in parentheses are performance improvements achieved by applying our method.

| Method | Untargeted attacks | | | Targeted attcks | | |
|---|---|---|---|---|---|---|
| | $PGD_{10}$ | $PGD_{50}$ | $PGD_{200}$ | $PGD_{10}$ | $PGD_{50}$ | $PGD_{200}$ |
| Mixup (Zhang et al., 2018) | 5.5 | 5.3 | 5.2 | $\leq 1$ | $\leq 1$ | $\leq 1$ |
| Ours + Mixup | 9.2 | 12.3 | 14.3 | 31.8 | 32.4 | 33.8 |
| Xie et al.'s (2018) + Mixup | 9.6 | 7.6 | 7.4 | 30.2 | 22.5 | 22.3 |
| Ours + Xie et al.'s + Mixup | $19.3_{(+9.7)}$ | $21.1_{(+13.5)}$ | $23.4_{(+16.0)}$ | $40.9_{(+10.7)}$ | $40.7_{(+18.2)}$ | $41.2_{(+18.9)}$ |
| Guo et al.'s (2018) + Mixup | 13.1 | 10.8 | 10.5 | 33.3 | 26.3 | 26.1 |
| Ours + Guo et al.'s + Mixup | $\mathbf{28.3}_{(+14.8)}$ | $\mathbf{30.9}_{(+20.1)}$ | $\mathbf{32.9}_{(+22.4)}$ | $\mathbf{50.8}_{(+17.5)}$ | $\mathbf{50.9}_{(+24.6)}$ | $\mathbf{51.7}_{(+25.6)}$ |
| MI-OL Pang et al. (2020) + Mixup | 12.6 | 9.4 | 9.1 | 37.0 | 29.0 | 28.7 |
| Ours + MI-OL + Mixup | $19.0_{(+6.4)}$ | $20.1_{(+10.6)}$ | $21.9_{(+12.8)}$ | $36.5_{(-0.5)}$ | $35.9_{(+6.9)}$ | $36.9_{(+8.2)}$ |
| IAT (Lamb et al., 2019) | 26.6 | 24.1 | 24.0 | 52.0 | 50.1 | 49.8 |
| Ours + IAT | 32.6 | 33.9 | 34.6 | 54.3 | 53.2 | 53.2 |
| Xie et al. (2018) + IAT | 42.2 | 41.5 | 41.3 | 57.1 | 56.3 | 55.8 |
| Ours + Xie et al.'s + IAT | $\mathbf{46.8}_{(+4.6)}$ | $\mathbf{48.1}_{(+6.6)}$ | $\mathbf{48.4}_{(+7.1)}$ | $\mathbf{58.7}_{(+1.6)}$ | $\mathbf{57.5}_{(+1.2)}$ | $\mathbf{57.4}_{(+1.6)}$ |
| Guo et al.'s (2018) + IAT | 36.2 | 33.7 | 33.3 | 53.8 | 52.4 | 52.2 |
| Ours + Guo et al.'s + IAT | $40.9_{(+4.7)}$ | $42.1_{(+8.4)}$ | $42.2_{(+8.9)}$ | $55.0_{(+1.2)}$ | $53.6_{(+1.2)}$ | $53.7_{(+1.5)}$ |
| MI-OL Pang et al. (2020) + IAT | 43.8 | 42.8 | 42.5 | 58.1 | 56.7 | 56.5 |
| Ours + MI-OL + IAT | $45.4_{(+1.6)}$ | $46.3_{(+3.5)}$ | $46.8_{(+4.3)}$ | $57.5_{(-0.6)}$ | $56.7_{(+0.0)}$ | $57.4_{(+0.9)}$ |

with non-predicted labels) (Pang et al., 2020). We compare the performance our method and the baselines and present results of the joint use of our method and the baselines to resist adversarial examples.

## 4.2 EXPERIMENTAL RESULTS

We validate the proposed method against oblivious-box attacks (Carlini & Wagner, 2017). That is the adversary does not know about the existence of the defense mechanism, and adversarial examples are generated only based on targeted network models. We evaluate the performance of defenses on the entire test set. Table 1 and Table 2 report the quantitative results on CIFAR-10 and CIFAR-100, respectively, demonstrating the effectiveness of the proposed method in improving defense performance. We see from Table 1 that the proposed method significantly helps to improve defense performance of the baseline methods against untageted attacks, achieving at least 12.5% and 4.1% performance gains for Mixup and IAT trained models, respectively. For defending against targeted attacks, the proposed method performs well in combination with Xie et al.'s and Guo et al.'s for Mixup trained models, and it performs well together with Xie et al.'s for IAT trained models. It can be seen from Table 2 that, as with on CIFAR-10, the proposed method also helps improve defense performance against untargeted attacks on CIFAR-100, achieving at least 6.4% and 1.6% performance improvements for Mixup and IAT trained models, respectively. For defending against targeted attacks, our method consistently helps to improve defense performance when applied on Xie et al.'s and Guo et al.'s methods. We can also make the following three observations from the quantitative results.

1. In most cases, the proposed method improves defense performance of the baseline methods. Especially for resisting untargeted attacks in different settings, our method significantly helps to improve defense performance. This shows that our method is complementary to the baselines to resist adversarial perturbations. Among the three baseline methods, the joint use of our method with Xie et al.'s and Guo et al.'s methods performs well compared to with the MI-OL method. This could be because the perturbation used to counteract adversarial perturbations is reduced due to the interpolation operation in MI-OL.

2. The proposed method performs well against strong PGD attacks with more iterations. Previous studies show that adversarial perturbations generated using more iterations are difficult to resist. The results of the baselines also show that PGD attacks with more iterations result in reduced performance. It is worth noting that the proposed method achieves improved performance for defending against most strong PDG attacks. And for the remaining attacks, the use of more iterations results in

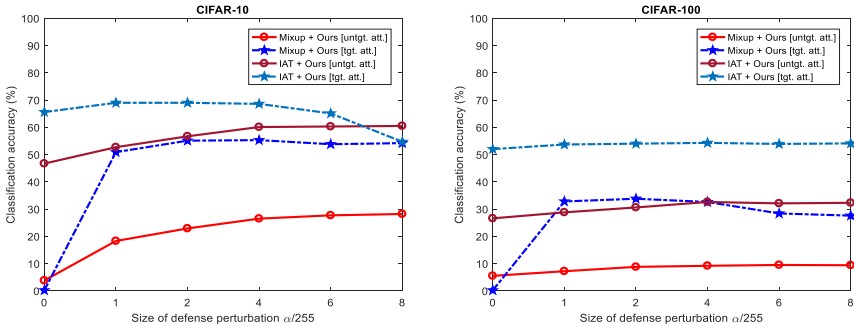

Figure 2: Impact of the size of perturbations generated for defense purposes on classification accuracy (%). We report performance of resisting both untargeted and targeted $PGD_{10}$ attacks.

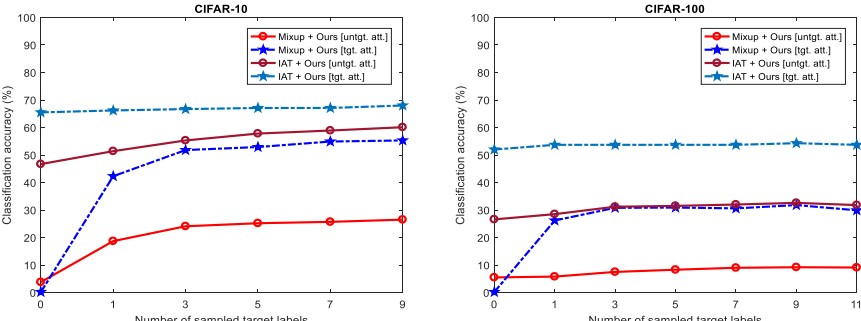

Figure 3: Impact of the number of randomly selected class labels in our method on classification accuracy (%). We report performance of resisting both untargeted and targeted $PGD_{10}$ attacks.

comparable performance as the use of less iterations. The results show that adversarial perturbations generated using more iterations can be easily counteracted by using first-order perturbations.

3. For defending against targeted $PGD_{50}$ and $PGD_{200}$ attacks on CIFAR-10, our method together with Guo et al.'s on Mixup trained models achieve higher performance than those obtained on IAT trained models, improving the classification accuracy 1.4% and 3.2%, respectively. Overall, our method together with Guo et al.'s achieve better or comparable performance than pure IAT trained models. As far as we know, we are the first to outperform pure adversarial training-obtained models using only inference time defense methods. This shows that it is promising that adversarial training could be unnecessary if proper perturbations are applied to adversarial examples.

Next, we analyse the impact of the step size used in the one-step PGD method on defense performance. We experiment on CIFAR-10 and CIFAR-100 resisting both untargeted and targeted $PGD_{10}$ attacks. The experimental results are reported in Figure 2. We see that the step size affects differently for untargeted and targeted attacks. The performance improves as the step size increases from 1 to 8 for untargeted tasks on the two datasets. For targeted attacks, the performance improves as the step size increases from 1 to 4 but starts to reduce or maintain similar performance as the step size further increases.

We also analyse the impact of the number of selected class labels in our method on defense performance. Figure 3 demonstrates the results of resisting untargetd and targeted $PGD_{10}$ attacks on CIFAR-10 and CIFAR-100. We see that the performance improves for both untargeted and targeted attacks as the number increases from 1 to 9 on CIFAR-10. On CIFAR-100, the performance also improves as the number increases from 1 to 9 but begins to drop or remain similar as the number further increases.

**Discussion on type of defense perturbations** In our experiments, small perturbations used to counteract the adversarial perturbation are generated in a targeted manner other than for targeted attacks on IAT trained models on CIFAR-100, small perturbations are generated in an untargeted manner. Overall, untargeted adversarial perturbations can be effectively counteracted using perturbations

Table 3: Classification accuracy (%) of different method used in combination with Guo et al.'s (2018). We report performance of resisting targeted attacks on Mixup trained models.

| Method | CIFAR-10 | | | CIFAR-100 | | |
|---|---|---|---|---|---|---|
| | $PGD_{10}$ | $PGD_{50}$ | $PGD_{200}$ | $PGD_{10}$ | $PGD_{50}$ | $PGD_{200}$ |
| Guo et al.'s (2018) | 57.8 | 49.1 | 48.9 | 33.3 | 26.3 | 26.1 |
| Gaussian noise    + Guo et al.'s | 63.9 | 63.8 | 63.3 | 34.0 | 30.3 | 29.0 |
| Random rotation  + Guo et al.'s | 63.8 | 63.6 | 63.2 | 40.8 | 36.4 | 35.1 |
| Xie et al.'s (2018) + Guo et al.'s | 59.1 | 58.3 | 58.2 | 41.1 | 35.8 | 33.7 |
| MI-OL (Pang et al., 2020) + Guo et al.'s | 62.9 | 62.6 | 65.1 | 37.6 | 34.0 | 35.7 |
| Ours             + Guo et al.'s | **74.1** | **78.0** | **79.8** | **50.8** | **50.9** | **51.7** |

generated in a targeted manner by our method. The results also suggest that adversarial training has an unstable behavior for different data distributions.

**Discussion on number of steps used to generate defense perturbations** The perturbations for defense purposes are generated using the one-step PGD method. We also experiment using multiple steps to generate perturbations for defense purposes. However, we find that this results in reduced performance in defending against adversarial examples. This could be because perturbations generated using multiple steps have adversarial effects and they do not help much to counteract the original adversarial perturbation.

Table 4: Classification accuracy (%) on clean examples.

| Method | CIFAR-10 | | CIFAR-100 | |
|---|---|---|---|---|
| | Mixup | IAT | Mixup | IAT |
| ResNet-50 (w/o defense) | 93.8 | 89.7 | 74.2 | 64.7 |
| Xie et al.'s (2018) | 82.1 | 82.1 | 66.3 | 62.1 |
| Guo et al's (2018) | 83.3 | 83.9 | 66.1 | 61.5 |
| MI-OL (Pang et al., 2020) | 83.9 | 84.2 | 68.8 | 62.0 |
| Ours (targted perturbations) | 61.2 | 75.3 | 8.3 | 47.4 |
| Ours (untargted perturbations) | 87.1 | 88.3 | 66.0 | 65.1 |

To demonstrate the advantage of our method, we further compare the performance of different methods used together with Guo et al.'s. The results of defending against attacks on Mixup trained models are reported in Table 3. We see that although these methods, including Xie et al.'s, MI-OL, as well as random rotation and Gaussian noise, are effective in improving performance, out methods outperforms these methods by a large margin, especially when resisting adversarial examples generated using more iterations.

Finally, we evaluate our method on clean examples. Table 4 compares the performance of our method and the baseline methods. We see that our method performs differently using different types of perturbations that are generated for defense purposes. Our method mostly performs very well on clean inputs compared to the baselines when the perturbations used for defense purposes are generated in an untargeted manner.

## 5 CONCLUSION

We proposed a method of counteracting adversarial perturbations for defending against adversarial attacks. In our method, we generate small perturbations for a number of randomly selected class labels and apply these small perturbations to the input to counteract the adversarial perturbation. Unlike previous methods, our method employs the use of local first-order gradient for defense purposes and can effectively improve adversarial robustness. Our method is applied at inference time and complementary to the adversarial training method to improve overall defense performance. We experimentally validated our method on CIFAR-10 and CIFAR-100 against both untargeted and targeted PGD attacks. We presented extensive results demonstrating our method significantly improves the defense performance of the baseline methods. We showed that our method performs well in resisting strong adversarial perturbations generated using more iterations, demonstrating the advantage of using local first-order gradient to resist adversarial perturbations. Notably, our method together with Guo et al.'s (2018) achieved better performance than those obtained on IAT trained models when resisting targeted $PGD_{50}$ and $PGD_{200}$ attacks. This shows that it is promising adversarial training could be unnecessary if proper perturbations are applied to inputs.

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

# A APPENDIX

## A.1 TRAINING USING MIXUP

In the Mixup method (Zhang et al., 2018), neural networks are trained by minimizing the following loss:

$$L(f) = \frac{1}{m} \sum_{i=1}^{m} \ell(f(\tilde{x}_i, \tilde{y}_i)), \tag{6}$$

where $\ell$ is a loss function that penalizes the difference between the prediction and its actual target, and

$$\tilde{x}_i = \lambda x_i + (1 - \lambda)x_j, \\ \tilde{y}_i = \lambda y_i + (1 - \lambda)y_j. \tag{7}$$

$(x_i, y_i)$ and $(x_j, y_j)$ are randomly sampled from the training data, $\lambda \sim \text{Beta}(\alpha, \alpha)$, $\alpha \in (0, +\infty)$. Training using Mixup empirically improves the generalization performance on clean samples and slightly improves robustness against adversarial examples.

## A.2 ADVERSARIAL TRAINING

Adversarial training was introduced by Szegedy et al. (2014). In the adversarial training method, a mixture of adversarial and clean examples are used train a neural network. Madry et al. (2018) formulated adversarially robust training of neural networks as the saddle point problem:

$$\min_{\theta} \rho(\theta), \text{ where } \rho(\theta) = \mathbb{E}_{(x,y)\sim\mathcal{D}} \left[ \max_{\delta \in \mathcal{S}} L(\theta, x + \delta, y) \right], \tag{8}$$

where $\theta$ denotes the parameters of the neural network and $\mathcal{S}$ is the allowed set for perturbations. The inner maximization problem aims to find an adversarial version of a given data point $x$ that achieves a high loss, while the outer minimization aims to find model parameters such that the adversarial loss given by the inner attack problem is minimized. PGD as a first-order adversary can reliably solve the inner maximization problem, even though the inner maximization is non-concave.

Lamb et al. (2019) proposed the interpolated adversarial training (IAT) method that combines Mixup with adversarial training. In the IAT method, the interpolation of adversarial examples and that of clean examples are used for training neural networks. Compared to adversarial training, IAT can achieve high accuracy on clean examples while maintaining adversarial robustness.

## A.3 MORE TECHNICAL DETAILS

The hyperparameter settings used in our method on CIFAR-10 and CIFAR-100 are given in Table 5 and Table 6, respectively.

Table 5: Parameter settings for experiments on CIFAR-10. For each experiment, the number of executions is set to 50 for Xie et al.'s and Guo et al.'s and to 30 for MI-OL. [T] denotes small perturbations are generated in a targeted manner and [U] denotes mall perturbations are generated in an untargeted manner.

| Method | Mixup models | IAT models |
|---|---|---|
| Ours + Xie et al.'s (2018) | Def. pertb. type: [T] Random crop size: [22 30] | Def. pertb. type: [T] Random crop size: [26 32] |
| Ours + Guo et al.'s (2018) | Def. pertb. type: [T] Random crop size: [22 30] | Def. pertb. type: [T] Random crop size: [24 32] |
| Ours + MI-OL (2020) | $\lambda_{OL}$=0.5 | $\lambda_{OL}$=0.6 |

Table 6: Parameter settings for experiments on CIFAR-100. For each experiment, the number of executions is set to 50 for Xie et al.'s and Guo et al.'s and to 30 for MI-OL. [T] denotes small perturbations are generated in a targeted manner and [U] denotes mall perturbations are generated in an untargeted manner.

| Method | Mixup models | IAT models |
|---|---|---|
| Ours + Xie et al.'s (2018) | Def. pertb. type: [T] Random crop size: [26 32] | Def. pertb. type: [T] Random crop size: [26 32] for untgt. att. and [24 32] for tgt. att. |
| Ours + Guo et al.'s (2018) | Def. pertb. type: [T] Random crop size: [24 32] | Def. pertb. type: [U] Random crop size: [24 32] |
| Ours + MI-OL (2020) | $\lambda_{OL}$=0.5 | $\lambda_{OL}$=0.6 |

