# OpenReview forum: "Towards Counteracting Adversarial Perturbations to Resist Adversarial Examples"
_ICLR.cc/2021/Conference — Reject_

### Official Review · AnonReviewer2 · 2020-10-26
**Evaluation is not rigorous**

**Rating:** 3
**Confidence:** 5

**Review:**

The author proposed a input transformation method for countering adversarial attacks. The proposed method can be applied to any pretrained network and improve the adversarial robustness.

However, I have some strong concerns regarding the evaluation method of this paper:

- The proposed method is based on the principle of obfuscated gradient, which has shown to be vulnerable against adaptive attacks [1]. The authors did not mention this at all in their paper, nor evaluate their method against adaptive attacks.
They only evaluate the adversarial accuracy under PGD attack. I believe this is based on the assumption that the attacker does not known the defense algorithm. However, this is not a comfortable assumption in the cryptography point of view when building a defense algorithm.
 The author should evaluate their method against some adaptive attacks before making the conclusion of whether the method is robust or not.

- As pointed out in [1], defenses based on obfuscated gradient may still suffer from gradient-free attacks. The author should also evaluate their method on gradient-free attacks like [2] or [3].

Other minor comments:

- Some baseline methods used by the author (e.g., Guo et al. and Xie et al.) have already been shown that are not effective [1]. It is not necessary to include them as baselines.

- This paper [4] also utilize the first-order gradient information at inference time. I'm not sure how similar is [4] to the proposed method but the author should discuss the difference in the related work.



[1] Anish Athalye, Nicholas Carlini, and David Wagner. "Obfuscated gradients give a false sense of
security: Circumventing defenses to adversarial examples." International Conference on Machine
Learning, 2018.

[2] Chen, Jianbo, Michael I. Jordan, and Martin J. Wainwright. "Hopskipjumpattack: A query-efficient decision-based attack." 2020 ieee symposium on security and privacy (sp). IEEE, 2020.

[3] Ilyas, Andrew, Logan Engstrom, Anish Athalye, and Jessy Lin. "Black-box Adversarial Attacks with Limited Queries and Information."  International Conference on Machine Learning, 2018.

[4] Xiao, Chang, and Changxi Zheng. "One Man's Trash Is Another Man's Treasure: Resisting Adversarial Examples by Adversarial Examples." Proceedings of the IEEE/CVF Conference on Computer Vision and Pattern Recognition. 2020.

---

### Official Review · AnonReviewer3 · 2020-10-27
**The defense evaluation violates many guidelines and best practices**

**Rating:** 2
**Confidence:** 5

**Review:**

The paper proposes a defense that works by adding multiple targeted adversarial perturbations (with random classes) on the input sample before classifying it. There is little theoretical reasoning for why this is a sensible defense. More importantly though, the defense is only evaluated in an oblivious threat model where the attacker is unaware of the defense mechanism. As has been argued again and again in the literature and in community guidelines such as [1, 2], the oblivious threat model is trivial and yields absolutely no insights into the effectiveness of a defense (e.g. you can just manipulate the backpropagated gradient in random ways to prevent any gradient-based attack from finding adversarial perturbations). The problem with oblivious attacks is clearly visible in the results section where more PGD iterations are less effective than fewer iterations - a clear red flag that the evaluation is ineffective. The paper also fails to point out that Pang et al. 2020, one of the methods they combine their method with, has been shown to be ineffective [2].

This work clearly violates the guidelines and best practices that the adversarial robustness community has developed and the claims are not substantiated by the evaluation or the results. Fixing this problem will require developing adaptive attacks that are effective against the proposed defense. [2] is a good starting point, in particular the attack against Pang et al. 2020. This will require time, care and effort to get right, and I don't see a way how reliable results from a new adaptive evaluation can be generated during the relatively brief rebuttal period.

[1] On Evaluating Adversarial Robustness, https://arxiv.org/abs/1902.06705
[2] On adaptive attacks to adversarial example defenses, https://arxiv.org/abs/2002.08347

---

### Official Review · AnonReviewer1 · 2020-10-28
**Threat model chosen is not justified, the defense can be broken using adaptive attacks**

**Rating:** 2
**Confidence:** 5

**Review:**

#################################### Summary ######################################

The paper presents an inference-time defense which generates single-step attacks towards other classes, and adds a sum of all these perturbations projected to an $\ell_\infty$ ball to the original image before inference. This combined with other existing methods achieves 69.3% robust accuracy on CIFAR-10 dataset. The authors claim to achieve a 11.4%  boost in robustness without the use of adversarial training.

##################################### Details #######################################

  -  The threat model chosen is not justified. The paper considers a white-box threat model where only the network model (architecture and weights) is accessible to the attacker. Even in the experiments section, only oblivious attacks which are unaware of the defense mechanism are considered. As explained by Carlini et al. [1], for white-box or black-box defenses, it is not reasonable to assume that the defense algorithm can be held secret. The only secret can be the randomness chosen during test time. In this case, the authors select a random subset of classes for the single-step attack during inference, and it can be assumed that the attacker is not aware of which classes are chosen. Apart from this, it must be assumed that all other details related to the defense are known to the attacker.  This is because an attacker is assumed to be infinitely thorough, without any computational restrictions, towards finding an adversarial perturbation within the threat model if it exists. The attacker could even use a naive attack which considers infinitely many random samples in the constraint set to fool the model. This random sampling based attack would almost surely succeed if an adversarial perturbation exists within the constraint set, and it also does not require knowledge of the defense. However, for practical evaluation, where we are limited by the computation available, we need to make use of the knowledge of the defense to find the strongest possible attack.
  -  While Carlini & Wagner define the Zero-Knowledge threat model in their earlier paper [2], they clarify in their subsequent work [1] that there is no justification for such a threat model, and it was defined only to highlight that some defenses were not even robust against such a weak threat model.
  -  Given that the defense must be known to the attacker, it is possible to construct an adaptive attack which includes the defense (addition of single-step attacks to multiple random classes) in each attack step. Different random classes can be chosen in each step of the attack. This is similar to the attack constructed by Tramer et al. [3] to break the defense by Pang et al. [4]. Using this attack, the accuracy can potentially be brought down to the baseline method (IAT). Another possible adaptive attack is EOT [5], since the proposed defense involves randomization.
  -  All the baselines and methods that the proposed defense is combined with, are broken in prior work. The work by Guo et al. [6] and Xie et al. [7] are broken using EOT and BPDA attacks by Athalye et al. [5]. The defense by Pang et al. [4] is broken by Tramer et al. [3]. Hence, they are not justified as baselines, and their contribution in the proposed combined defense can also be nullified using the same attacks.
  -  Robustness evaluation has only been done using PGD-200. Robust accuracy needs to be reported on more recent attacks such as AutoAttack [8] and Multi Targeted attack [9]. As mentioned above, even for these evaluations, the defense needs to be considered while generating the attack.

[1] Carlini et al., On evaluating Adversarial Robustness, https://arxiv.org/abs/1902.06705

[2] Adversarial Examples Are Not Easily Detected: Bypassing Ten Detection Methods, ACM Workshop, 2017

[3] Tramer et al., On Adaptive Attacks to Adversarial Example Defenses, NeurIPS 2020

[4] Pang et al., Mixup inference: Better exploiting mixup to defend adversarial attacks, ICLR 2020

[5] Athalye et al., Obfuscated Gradients Give a False Sense of Security: Circumventing Defenses to Adversarial Examples, ICML 2018

[6] Guo et al., Countering adversarial images using input transformations, ICLR 2018

[7] Xie et al., Mitigating adversarial effects through randomization. ICLR 2018.

[8] Croce et al., Reliable Evaluation of Adversarial Robustness with an Ensemble of Diverse Parameter-free Attacks, ICML 2020

[9] Gowal et al., An Alternative Surrogate Loss for PGD-based Adversarial Testing, https://arxiv.org/pdf/1910.09338.pdf

################################# Reasons for the score #################################

The paper rests on a threat model that considers the attacker to be oblivious to the defense used. Prior works [1, 3] consider such a threat model to be unjustified. An adaptive adversary cognizant of the defense can potentially break the proposed defense by incorporating the defense during attack generation, and by using EOT attacks. Hence I strongly vote to reject the paper.

---

### Official Review · AnonReviewer4 · 2020-10-30
**Review for Towards Counteracting Adversarial Perturbations to Resist Adversarial Examples**

**Rating:** 1
**Confidence:** 5

**Review:**

The paper proposes a defense against adversarial examples. The idea of the defense is to counteract transformation generated by PGD attack. This is done by running one step of PGD on network input several times (and using different target labels) and then average the result.

As described below, I think there are multiple serious flaws in the evaluation and the defense likely won’t work. Thus I recommend rejecting the paper.

Issues with the paper:
* Experiment section is lacking rigor which is necessary to properly evaluate defense against adversarial examples. Points below explain it in more detail.
* One clear indication of a problem with evaluation is the fact that accuracy under attack is increasing (see table 2) when seemingly stronger attack is used. [i.e. PGD with larger number of steps]
* The whole premise of the defense idea is to counteract very specific thing which PGD does, thus it’s unlikely to help against more sophisticated attacks or simply different attacks.
* Evaluation procedure implies that the attacker has no knowledge about the defense and even no ability to query the defended model. Which is very strong restrictions on the attacker, moreover they are impractical from security standpoint [even in black box case attacker usually has an ability to query the model]
* Authors cite (Carlini & Wagner, 2017) to justify oblivious-box attack setup. However  (Carlini & Wagner, 2017) actually proposes white-box attack and provides justification why white-box should be used.
* https://arxiv.org/pdf/1802.00420.pdf (which authors cite) shows how to break defenses based on input transformations. Nevertheless authors do not address why the proposed defense (which is also input transformation) is not broken.
* Authors use Guo et al.’s (2018) as one of the baselines, despite that this is a broken defense (per https://arxiv.org/pdf/1802.00420.pdf )
* No comparison with adversarial training, in particular authors should consider comparing to  A. Madry’s paper https://arxiv.org/pdf/1706.06083.pdf which also performs experiments on CIFAR dataset.


Feedback on how to improve paper:
* The evaluation should be performed in the assumption that adversary is either aware of the attack (white-box case) or able to query the defended model (black box attack).
* Use https://arxiv.org/abs/1902.06705 as a guide on how to properly evaluate models for adversarial robustness. And redo evaluation following this guide.
* In particular, consider running gradient free attacks on the whole model (baseline model + defense on top of it) and try to make an attack which will break the proposed defense.

---

> ### Public Comment · ~Nicholas_Carlini1 · 2020-11-10
> **I agree that "oblivious box" is bad**
>
> I wrote (Carlini & Wagner, 2017). We made a mistake in this paper by making it look like oblivious attacks matter---they do not. As we said in the follow-up paper ("On evaluating adversarial robustness"):
>
> Along the same lines, there is no justification to study a “zero-knowledge” (Biggio et al.,
> 2013) threat model where the attacker is not aware of the defense. “Defending” against such
> an adversary is an absolute bare-minimum that in no way suggests a defense will be effective
> to further attacks. Carlini & Wagner (2017a) considered this scenario only to demonstrate
> that some defenses were completely ineffective even against this very weak threat model.
> The authors of that work now regret not making this explicit and discourage future work
> from citing this paper in support of the zero-knowledge threat model.

---

### Decision · Program_Chairs · 2021-01-07
**Final Decision**

**Decision:**

Reject

**Comment:**

This work is attempting to develop a new way to train models that are robust to (l_p-bounded) adversarial perturbations and to do so in a way that departs from the tools successfully used for this purpose in the past. This is a worthwhile aspiration, however, as pointed out in the comments/reviews, there are significant problems with the methodology of evaluating the proposed approach (and some well-founded skepticism that this approach is indeed successful). As such, this paper cannot be accepted in its current form.